# ZERO-SHOT TASK-LEVEL ADAPTATION VIA COARSE-TO-FINE POLICY REFINEMENT AND HOLISTIC-LOCAL CONTRASTIVE REPRESENTATIONS

## ABSTRACT

Meta-reinforcement learning offers a mechanism for zero-shot adaptation, enabling agents to handle new tasks with parametric variation in real-world environments. However, existing methods still struggle with task-level adaptation, which demands generalization beyond simple variations within tasks, thereby limiting their practical effectiveness. This limitation stems from several challenges, including the poor task representations and inefficient policy learning, resulting from the underutilization of hierarchical structure inherent in task-level adaptation. To address these challenges, we propose a Coarse-to-Fine Policy Refinement combined with a Holistic-Local Contrastive Representation method to enable effective zero-shot policy adaptation. Specifically, in terms of policy learning, we use task language instructions as prior knowledge to select skill-specific expert modules as a coarse policy. This coarse policy is then refined by a fine policy generated through a hypernetwork, producing a task-aware policy based on task representations. Additionally, for task representation, we employ contrastive learning from both holistic and local perspectives to enhance task representations for more effective policy adaptation. Experimental results demonstrate that our method significantly improves learning efficiency and zero-shot adaptation on new tasks, outperforming previous methods by approximately $42.3\%$ and $45.4\%$ in success rate on the Meta-World ML-10 and ML-45 benchmarks, respectively.

## 1 INTRODUCTION

The dynamic and unpredictable nature of the real world presents significant challenges for agents operating within it. Improving agents' adaptability in such environments is essential, as their performance hinges on effectively managing these changes. Zero-shot adaptation (Shinzaki et al., 2021; Ball et al., 2021) represents an ideal form of adaptability, allowing agents to excel in new tasks from the first episode without pre-collecting samples or updating network parameters. However, traditional reinforcement learning (RL) methods typically do not endow agents with the ability. These methods are usually tailored to specific tasks, requiring agents to learn from scratch for each new task, which is inefficient in real-world scenarios.

Context-based meta-reinforcement learning offers a promising approach for improving agents' zero-shot adaptation to unseen tasks. This method involves task representation and policy execution. It first infers task representations from contextual information and then adjusts the policy based on these representations and the environmental state. However, most existing methods are often meta-trained on narrow task distributions, where different tasks are merely defined by varying a few parameters that specify the reward function or environment dynamics. This process is referred to as variation-level adaptation, as illustrated in Figure 1a. Although the relationship between such tasks is well-defined, agents gain limited inductive bias from the narrow distribution, leading to difficulties in generalizing to new tasks with greater diversity, namely task-level adaptation (Zhao et al., 2022; Team et al., 2024), is illustrated in Figure 1b. Task-level adaptation has two hierarchical interpretations. The first involves the presence of shared subtasks across different task categories. These subtasks represent skills that can be reused across multiple tasks, making them common to a variety of task types. The second interpretation involves two distinct levels of adaptation: at the higher level, the agent adapts to new tasks across various categories; at the lower level, the agent

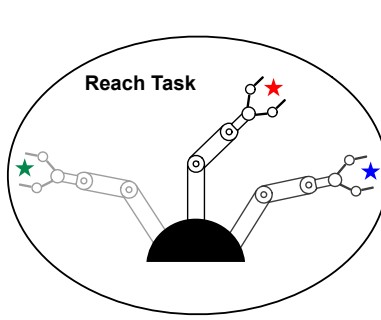 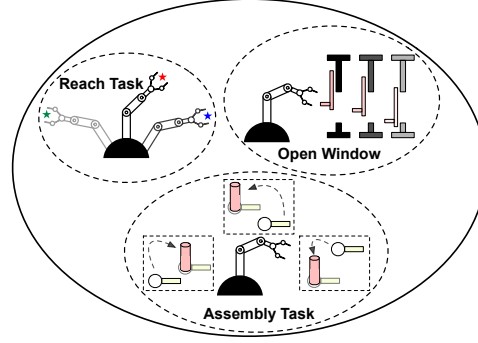

(a) Variation-level Adaptation        (b) Task-level Adaptation

Figure 1: **Variation-level Adaptation vs. Task-level Adaptation**. Variation-level adaptation refers to changes that occur within the scope of specific tasks. In contrast, task-level adaptation requires the agent to adapt not only across multiple task categories but also to different variations of tasks within specific task categories.

adapt to different instances of tasks within a single category. Consequently, agents need to adapt both across different task categories and within task variations inside a category, posing a significant challenge to existing approaches. Moreover, because task-level adaptation more accurately reflects real-world environments, it is crucial for agents to manage this adaptation effectively.

Several existing approaches have been proposed to address task-level adaptation. SDVT (Lee et al., 2023) utilizes a Gaussian Mixture VAE to meta-learn the task decomposition process, incorporating a virtual training procedure to enhance generalization to previously unseen tasks. Meanwhile, Million (Bing et al., 2023) integrates transformers with task language instruction to improve task adaptation capabilities. While these methods improve adaptation to new tasks, they encounter certain challenges. Although SDVT implicitly introduces hierarchy into task representation by using a Gaussian mixture VAE to model the latent space, the task representations obtained through SDVT may not generalize well to unseen task categories because it fails to directly constrain task representations from the perspectives of both different task categories and individual task instances within a category, thereby reducing its robustness. Additionally, it does not explicitly incorporate hierarchy into policy execution, thereby failing to leverage shared skills across different task categories, which results in suboptimal adaptation performance. Conversely, the Million method demands a large volume of training data due to its reliance on the transformer architecture, which makes it impractical for online paradigms. Furthermore, like SDVT, it suffers from learning inefficiencies as it does not effectively leverage the hierarchy inherent in task-level adaptation. Consequently, our intuition is that introducing hierarchical characteristics of task-level adaptation into task representation and policy learning can enhance task adaptation performance.

In this paper, we present a novel framework for meta-RL that incorporates **C**oarse-to-**F**ine p**O**licy refinement with a **H**olistic-**L**ocal contrastive task **R**epresentation (**CFOHLR**). It utilizes a context-based meta-RL architecture comprising a task inference module and a conditional policy module. Based on our intuition, our method is grounded on two key insights. First, effective task-level adaptation requires an agent to have a general understanding of task forms and to select appropriate skills accordingly. To achieve this, we employ language instructions to provide the agent with the necessary comprehension of the tasks. We establish multiple skill-specific expert networks, which are selected based on these instructions, forming the coarse policy level. However, since different task attributes can further influence performance, the agent also needs a fine-grained, task-aware policy that adapts to the specific attributes of each task. Therefore, we utilize a hypernetwork to generate this policy based on task attributes, forming the fine policy. By combining language-guided expert skill selection with a hypernetwork-based task-aware policy, we achieve a coarse-to-fine policy refinement. Second, developing an effective task-aware policy depends on accurately capturing task attributes through robust task representations. To achieve this, we propose a holistic-local contrastive task representation method. This approach is based on the insight that task representations should first be distinctly separated at the task category level, and then further differentiated among tasks within the same category. Specifically, we employ contrastive learning to enforce that task

representations are distinctly separated in the representation space. This approach refines task representations from both holistic and local perspectives, where the holistic view corresponds to general task categories and the local view addresses task category-specific instances. Consequently, this results in more robust and informative task representations for generating task-aware policy.

We evaluate our proposed method on the Meta-World ML10 and ML45 benchmarks, which are widely used to assess task-level adaptation performance across diverse robotic manipulation tasks. The experimental results demonstrate that our method significantly enhances both learning efficiency and zero-shot adaptation capabilities in new tasks, outperforming previous meta-RL approaches. In summary, our contributions are as follows:

• We propose a coarse-to-fine policy refinement that integrates language-guided expert skill selection as the coarse policy with a hypernetwork-based task-aware policy as the fine policy, enhancing learning efficiency and zero-shot adaptation to new tasks.

• We introduce a holistic-local contrastive task representation at both the general task category level and the task category-specific instance level to enhance the robustness of task representations, thereby enabling the generation of task-aware policy.

• We conduct extensive experiments on the Meta-World benchmarks to validate the effectiveness of our method, outperforming previous methods by approximately $42.3\%$ and $45.4\%$ in success rate on the Meta-World ML-10 and ML-45 benchmarks, respectively.

## 2 PRELIMINARY

### 2.1 META-REINFORCEMENT LEARNING

In traditional RL, most problems are typically formalized as Markov Decision Processes (MDPs) (Bellman, 1966). An MDP is defined as a tuple $M = (S, A, P, \rho_0, R, \gamma)$, where $S$ represents the state space, $A$ denotes the action space, $P(s'|s, a)$ is the transition function, $\rho_0(s)$ is the initial state distribution, $R(s, a)$ is the reward function, and $\gamma$ is the discount factor. The objective of RL is to maximize the expected cumulative reward $J(\pi) = \mathbb{E}_\tau \left[ \sum_{t=0}^{\infty} \gamma^t R(s_t, a_t) \right]$ in order to obtain an optimal policy $\pi$.

When extending RL to meta-RL, a distribution of MDPs is introduced, denoted as $p(M)$, where each MDP is characterized by distinct reward or transition dynamic functions. MDPs sampled from this distribution represent individual tasks that share the same state and action spaces but differ in their respective reward or transition dynamics functions. Meta-RL utilizes meta-knowledge acquired from prior training tasks to aid agents in tackling new tasks. Notably, in contrast to multi-task RL, the agent in meta-RL does not have access to explicit task-related information; instead, it must infer task attributes through interaction with the environment. Meta-RL aims to maximize the expected cumulative rewards across the training task distribution to obtain optimal policy $\pi_\theta$:

$$J(\pi_\theta) = \mathbb{E}_{M \sim p(M)}[J_M(\pi_\theta)]. \tag{1}$$

### 2.2 TASK ATTRIBUTES INFERENCE

In the process of adapting to new tasks, an agent must gather contextual information through interactions with the environment to infer task attributes and adjust its policy accordingly to maximize returns. Regarding task inference, two primary methods currently exist to infer task attributes. The first method is posterior sampling-based, where the agent samples a single hypothesis MDP from its posterior distribution. The agent then follows the optimal policy for the sampled MDP until the next sample is drawn, repeating this process to update the posterior distribution. The second method is based on the Bayesian Adaptive MDP (BAMDP) (Duff, 2002). The BAMDP-based method is preferred because it effectively balances exploration (collecting trajectory information that reflects task attributes) and exploitation (reasoning about task attributes based on the collected trajectory information), thereby offering greater efficiency. VariBAD (Zintgraf et al., 2019) employs the BAMDP framework by meta-training a Variational Auto-Encoder (VAE) (Kingma & Welling, 2013) to extract task representations from historical trajectories. Similarly, SDVT (Lee et al., 2023) adopts a comparable approach but distinguishes itself by using a Gaussian mixture distribution to model the

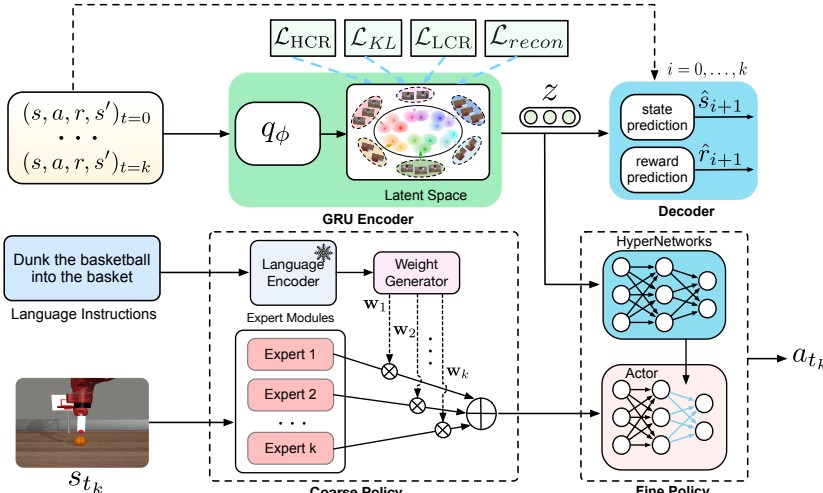

Figure 2: **CFOHLR architecture.** Our framework comprises two modules: task inference and policy execution. In the task inference module, the encoder first extracts a task representation, $z$, from an online consecutive trajectory. Simultaneously, the decoder predicts states and rewards to compute the reconstruction loss. In the policy execution module, language instructions are utilized to select skill-specific expert modules as a coarse policy, which is then refined by a fine policy. The fine policy employs a hypernetwork to generate a task-aware policy based on the task representation.

latent space. This method is particularly well-suited for handling complex tasks. The VAE consists of an encoder, $q_\phi(m|\tau_{:t})$, which generates task representations, and a decoder that forecasts future rewards and states, contributing to the reconstruction loss used during meta-training. The training objective of SDVT is

$$\mathcal{L}_{\text{VAE}}(\phi,\theta) = \mathbb{E}_{p(M)}\left[\sum_{t=0}^{H^+} ELBO_t(\phi,\theta)\right] = \mathcal{L}_{\text{recon}} + \mathcal{L}_{\text{KL}}, \tag{2}$$

where

$$\begin{aligned} ELBO_t = &\mathbb{E}_{p(M)}\left[\mathbb{E}_{q_\phi(m|\tau_{:t})}\left[\log p_\theta\left(\tau_{:H^+}\mid m\right)\right] \\ &- KL\left(q_\phi\left(m\mid \tau_{:t}, y_t\right)\|q_\phi\left(m\mid y_t\right)\right)\right], \end{aligned} \tag{3}$$

$H^+$ is the horizon in the BAMDP, $y_t$ represents the mixture proportion of the current task among different tasks. The objective is to maximize evidence lower bound (ELBO), comprising a reconstruction term for the trajectory and a KL divergence relative to the previous posterior.

Similarly, we adopt this method to generate task representation at the current time step, utilizing historical information up to this point. In contrast, we utilize the hierarchical characteristics inherent in task-level adaptation to enhance the robustness of the task representation.

## 2.3 VARIATION-LEVEL ADAPTATION AND TASK-LEVEL ADAPTATION

The current meta-RL community typically evaluates algorithms using variations of the same training tasks, such as modifying dynamic functions (e.g., adjusting friction parameters) or altering reward functions (e.g., setting different target velocities). These evaluations fall under the category of variation adaptation. However, variation adaptation alone does not fully assess the effectiveness of meta-RL algorithms and is not entirely applicable to real-world scenarios. A more challenging form of adaptation is task-level adaptation, which involves training on a wide variety of tasks and generalizing to entirely novel tasks during testing. For instance, in the ML10 benchmark of MetaWorld, an agent might be trained on tasks such as pressing a button, closing a drawer, and picking and placing objects. However, during testing, the agent's ability to adapt would be evaluated on entirely unseen tasks, such as pulling a lever or placing an object on a shelf.

## 3 METHOD

This section introduces our framework, which integrates the joint training of the task inference module and the conditional policy module in an online setting. Our framework is designed to facilitate efficient task-level adaptation. We begin with an overview of our proposed method in Sec.3.1. Subsequently, we detail the coarse-to-fine policy refinement in Sec.3.2, followed by the holistic-local contrastive task representations in Sec.3.3.

### 3.1 METHOD OVERVIEW

Our proposed framework consists of two key components: a coarse-to-fine policy refinement and holistic-local contrastive task representations. As depicted in Figure 2, our method leverage the hierarchical characteristics inherent in task-level adaptation to enhance adaptation performance. To achieve this, we introduce a coarse-to-fine policy refinement, which integrates a language-guided mechanism for selecting specific-skill experts as the coarse policy with a hypernetwork-based task-aware policy as the fine policy. Additionally, to develop robust and generalizable task representations for generating task-aware policies, we introduce holistic-local contrastive task representations that operate at both the task category level and the task category-specific instance level.

### 3.2 COARSE-TO-FINE POLICY REFINEMENT.

Achieving superior performance in task-level adaptation requires an agent to possess a foundational understanding of the task's general structure and to refine this understanding through interactions with the environment. Similar to how humans execute tasks, individuals typically begin with an initial comprehension of the overall task and the approximate skills needed for completion. This understanding is progressively deepened through continuous interaction, enabling a more nuanced grasp of the task's attributes. To emulate this human-like task execution process, we propose a method that leverages language instructions to provide an initial understanding of the task, which is subsequently refined through interactions with the environment. Specifically, our approach begins by using language instructions to select a set of skill-specific expert modules, forming a coarse policy that captures the general outline of the required actions. This coarse policy is then refined by a subsequent stage that adapts the policy based on interactions with the environment. For implementation, we have developed multiple skill-specific expert modules. Language instructions are used to softly select among these experts, effectively composing the coarse policy. The output from the coarse policy is then fed into a refinement stage that employs a hypernetwork to generate a task-adaptive policy. This hypernetwork adjusts the policy parameters in response to task-specific attributes observed during interaction, enabling the agent to fine-tune its actions and achieve superior performance.

**Coarse Policy.** To design a coarse policy for the language-guided selection of skill-specific experts, we employ a fixed pre-trained DistilBERT sentence encoder (Sanh et al., 2019) to encode natural language task descriptions into fixed-length vectors in $\mathbb{R}^{768}$. The encoded vector, denoted as $z_{\text{instr}}$, is then used as input to an expert weight generation network, which outputs the weights $\alpha_1, \ldots, \alpha_k$ for the skill-specific expert modules. This process is formalized as follows:

$$\alpha_1, \ldots, \alpha_k = \text{softmax}\left(\mathcal{W}\left(z_{\text{instr}}\right)\right), \quad (4)$$

where $\mathcal{W}$ is a fully connected layer, and $\text{softmax}$ ensures that the weights $\alpha_i$ sum to 1.

The final coarse policy, denoted as $\pi_{\text{coarse}}$, is computed as a weighted sum of the $k$ expert-specific policy modules, where each expert policy is represented by $\pi_{\text{expert}}^j$. The weights $\alpha_j$, derived from the attention mechanism, determine the contribution of each expert policy:

$$\pi_{\text{coarse}} = \sum_{j=1}^{k} \alpha_j \cdot \pi_{\text{expert}}^j. \quad (5)$$

This formulation allows the coarse policy to combine multiple skill-specific expert policies based on the task instruction.

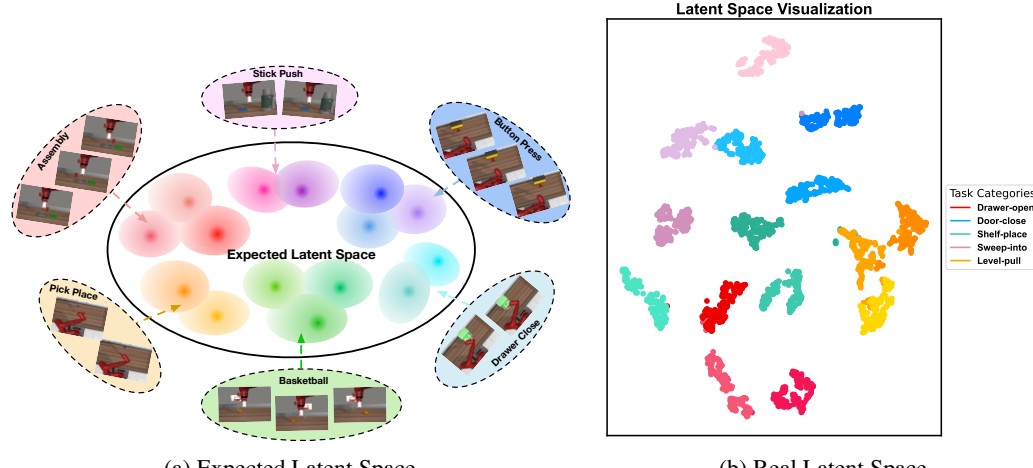

(a) Expected Latent Space                    (b) Real Latent Space

Figure 3: **Expected Latent Space and Real Latent Space**. **Left:** Our intuition is that task representations should first be distinctly separated at the task category level and then further differentiated among tasks within the same category. **Right:** The t-SNE visualization of the learned task representation space for the ML-10 testing tasks is presented. We sampled three tasks from each task category of the test tasks, with each color scheme representing a different task category. Each point in the visualization corresponds to a task representation vector extracted from transitions and is color-coded according to the task properties.

**Fine policy.** While language instructions guide the initial selection of relevant skill-specific expert modules, relying solely on the coarse policy may be insufficient for tasks in environments with dynamic attributes like varying object positions. To address this, we capture a task representation $z_t$ that reflects these environmental attributes and refine the policy accordingly. Specifically, followed by R2PGO (Li et al., 2024), we employ a hypernetwork $\mathcal{H}$ to generate a task-aware control policy $\pi_{\mathcal{H}(z_t)}$ in real time, based on the task representation obtained through a task inference module. This fine policy enhances the agent's ability to adapt to different task attributes.

In summary, to improve the agent's performance in task-level adaptation, we combine the strengths of both the coarse and fine policies. We first use language instructions to select and weight the skill-specific expert modules, forming the coarse policy. We then refine this policy using the task-aware control policy generated by the hypernetwork $\mathcal{H}$, which takes the task representation $z_t$ as input.

### 3.3 HOLISTIC-LOCAL CONTRASTIVE REPRESENTATION

While the coarse-to-fine control framework enhances task adaptation performance, generating a task-aware policy heavily relies on robust and generalizable task representations. Therefore, it is essential to develop a method to produce these robust representations. In task-level adaptation, one inevitably encounters various task categories, as well as multiple task instances within each category. For example, in the push task of the Meta-World benchmark, pushing items to different goal locations within the push task category can be considered as distinct task instances. Inspired by the hierarchical characteristics of task-level adaptation, we propose that task representations should capture both inter-category distinctiveness and intra-category differentiation, as shown in Fig 3a. To achieve this, we employ contrastive learning to derive robust task representations from two perspectives: the holistic, which addresses the task category level, and the local, which focuses on the task category-specific instance level. The real latent space is visualized in Fig 3b.

**Holistic Contrastive Representation.** From a holistic perspective, our focus is on the task category level, aiming to make task representations from different categories distinguishable. While contrastive learning is typically employed to obtain robust representations at the instance level within specific task categories (Li et al., 2021; Wang et al., 2023), we apply it at the task category level to achieve robust and discriminative representations across categories.

To reduce computational complexity, we represent each task category by averaging the task representations within that category. Specifically, we compute the task representations $c_i$ for each task type $i$ by averaging the representations of all tasks within that category. The formula is expressed as follows:

$$c_i = \frac{1}{N_i} \sum_{n=1}^{N_i} c_i^n, \qquad (6)$$

here, $N_i$ denotes the number of tasks associated with category $i$. Given a query task representation vector $z$ from task category $i$, we treat the pair $(z, c_i)$ as a positive pair. The averaged task representations from the remaining task categories serve as negative samples. The objective function for holistic contrastive representations, denoted as $\mathcal{L}_{\text{HCR}}$, is then defined as follows:

$$\mathcal{L}_{\text{HCR}} = -\frac{1}{N_{\text{category}}} \sum_{i=1}^{N_{\text{category}}} \log \left[ \frac{\exp\left(c_i \cdot x_i^+/\tau\right)}{\sum\limits_{j=1, j\neq i}^{N_{\text{category}}} \sum\limits_{k=1}^{N_j^-} \exp\left(c_i \cdot x_{ijk}^-/\tau\right)} \right], \qquad (7)$$

here, $N_{\text{category}}$ represents the number of task categories, $N_{j-}$ denotes the total number of negative samples corresponding to a specific task category, $x_i^+$ is the positive sample for task category $i$, and $x_{ijk}^-$ is the $k$-th negative sample from task category $j$ corresponding to task category $i$.

**Local Contrastive Representation.** From a local perspective, our focus is on task category-specific instance levels. Within a given task category, we aim for representations of the same task to be closely clustered, while representations of different tasks remain distinct. To achieve this structure, we apply contrastive learning to shape the latent space of task representations.

Specifically, for a given task category, we designate the task representation $z_t$ at a particular timestep as the query sample $x$ and select the task representation from the same task at a different time step as the positive sample $x^+$. Task representations from other tasks within the same category serve as negative samples $\{x_i^-\}_{i=1}^{N-1}$. Accordingly, we define the objective function for local contrastive representations, denoted as $\mathcal{L}_{\text{LCR}}$, is then defined as follows:

$$\mathcal{L}_{\text{LCR}} = -\frac{1}{N_{\text{category}}} \sum_{i=1}^{N_{\text{category}}} \frac{1}{N_{\text{tasks}}} \sum_{j=1}^{N_{\text{tasks}}} \log \left[ \frac{\exp\left(x_{ij} \cdot x_{ij}^+/\tau\right)}{\sum\limits_{k=1, k\neq j}^{N_{\text{tasks}}} \exp\left(x_{ij} \cdot x_{ijk}^-/\tau\right)} \right], \qquad (8)$$

where $N_{\text{category}}$ represents the number of task categories, $N_{\text{tasks}}$ denotes the total number of sampled tasks, $x_{ij}^+$ is the positive sample corresponding to the query sample $x_{ij}$, and $x_{ijk}^-$ is the $k$-th negative sample from task $j$ corresponding to task $i$. Consequently, we adopt a composite loss function that combines reconstruction and contrastive learning objectives: $\mathcal{L}_{\text{task inference}} = \mathcal{L}_{\text{VAE}} + \lambda_{\text{HCR}} \cdot \mathcal{L}_{\text{HCR}} + \lambda_{\text{LCR}} \cdot \mathcal{L}_{\text{LCR}}$.

## 4 EXPERIMENTS

### 4.1 EXPERIMENTAL SETTINGS

**Environments.** We evaluate our proposed method using the Meta-World benchmarks (Yu et al., 2020), which assess the generalization capabilities of agents across a wide range of task distributions. This benchmark contains 50 qualitatively distinct robotic manipulation tasks, each with 50 parametric variants that incorporate randomized goals and initial object positions. Specifically, the Meta-Learning 10 (ML-10) benchmark consists of $N_{\text{train}} = 10$ training tasks and $N_{\text{test}} = 5$ test tasks. Likewise, the Meta-Learning 45 (ML-45) benchmark comprises $N_{\text{train}} = 45$ training tasks and $N_{\text{test}} = 5$ test tasks. Notably, task IDs are not provided as input; agents need to identify task attributes from experience while maximizing their return within a meta-episode of $H^+ = 1000$ steps, which consists of $n_{\text{roll}} = 2$ rollout episodes of horizon $H = 500$ steps each.

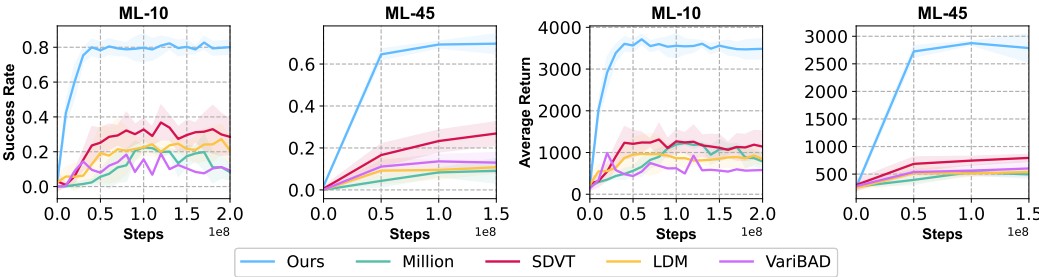

Figure 4: **Meta-World Success Rates and Returns in Test Tasks.** The success rates and corresponding average returns of our methods and baselines, averaged across the test tasks of ML-10 and ML-45 in the second rollout, are presented. The individual maximum success rates and corresponding returns for all tasks are reported in Appendix D.1.

**Baselines.** To demonstrate the effectiveness of our method, we compare it with the following methods: (1) **VariBAD** (Zintgraf et al., 2019) leverages a VAE consisting of an RNN-based encoder and a prediction decoder as a task inference module to obtain task representations, which are then used for decision-making. (2) **LDM** (Lee & Chung, 2021) utilizes synthetic tasks generated from mixtures of learned latent dynamics to enhance the generalization ability of agents. (3) **SDVT** (Lee et al., 2023) employs a Gaussian mixture VAE to meta-learn the task decomposition process and leverages a virtual training procedure to enhance generalization to unseen tasks. (4) **Million** (Bing et al., 2023) introduces a meta-RL paradigm comprising an instruction phase and a trial phase, integrating transformers with language instructions to improve task adaptation capabilities. To guarantee a fair comparison, each method is evaluated under the same experimental settings.

## 4.2 Comparison Unseen Tasks Adaptation Performance

To evaluate the performance of our method, we compare it with other approaches. In Figure 4, we present the mean and standard deviation of returns and success rates across five random seeds. Performance is assessed based on the success rate and average return across all test tasks.

Figure 4 illustrates that our method outperforms other baselines in both the ML-10 and ML-45 tasks. This success can be attributed to two central aspects of our approach: First, we implement a coarse-to-fine policy refinement strategy, allowing the agent to initially utilize skill-specific expert modules, followed by further refinement using a task-aware policy informed by task representations. Second, we apply contrastive learning to structure the latent space at both holistic and local levels, thereby producing more robust task representations for generating task-aware policies. As a result, our methodology significantly improves learning efficiency and adaptive performance, representing an advancement over previous state-of-the-art approaches.

## 4.3 Comparison Zero-shot Adaptation Performance

To evaluate the zero-shot adaptation performance of our method, we compared it with other approaches on the ML10 and ML45 tasks during the initial episodes.

Table 1 demonstrates that our method achieves respectable performance within the first episode when adapting to new tasks, outperforming other baselines across all environments. This demonstrates the strong zero-shot adaptation capabilities of our method, which are essential for agents functioning in dynamic and open-ended environments. While methods such as SDVT and LDM exhibit relatively good performance, they do not attain the highest performance in the first episode.

## 4.4 Ablation

To validate each proposed component of our method, we conducted a series of ablation experiments. The coarse-to-fine policy refinement and the holistic-local contrastive task representations are crucial elements of our approach. We compared our method with variants that excluded either the coarse-to-fine policy refinement or the holistic-local contrastive task representations to evaluate

Table 1: **Success Rate and Return on ML-10 and ML-45 Benchmarks.** To demonstrate their adaptability to unseen tasks, the meta-trained policies were rolled out over two episodes. We present the maximum success rate averaged across five random seeds, along with the corresponding returns.

| | ML-10 | | ML-45 | |
|---|---|---|---|---|
| **Methods** | **Episode 1** | **Episode 2** | **Episode 1** | **Episode 2** |
| | **Success Rate (%)** | | | |
| Ours | $83.9 \pm 2.9$ | $\mathbf{85.7 \pm 4.9}$ | $\mathbf{72.4 \pm 3.0}$ | $71.4 \pm 3.5$ |
| VariBAD | $23.0 \pm 10.9$ | $25.7 \pm 7.5$ | $13.8 \pm 4.4$ | $15.0 \pm 6.5$ |
| LDM | $34.6 \pm 17.6$ | $35.4 \pm 17.1$ | $11.6 \pm 5.5$ | $13.2 \pm 6.0$ |
| SDVT | $41.6 \pm 11.0$ | $43.4 \pm 9.4$ | $24.9 \pm 7.6$ | $27.0 \pm 8.9$ |
| Million | $25.1 \pm 7.7$ | $25.8 \pm 9.1$ | $11.5 \pm 7.2$ | $11.6 \pm 7.1$ |
| | **Return** | | | |
| Ours | $3697.5 \pm 201.9$ | $\mathbf{3733.5 \pm 164.9}$ | $\mathbf{2925.0 \pm 82.2}$ | $2893.3 \pm 107.5$ |
| VariBAD | $864.4 \pm 244.3$ | $929.1 \pm 208.1$ | $545.6 \pm 89.0$ | $631.0 \pm 202.8$ |
| LDM | $1173.3 \pm 723.7$ | $1151.3 \pm 692.0$ | $507.6 \pm 156.9$ | $597.2 \pm 153.7$ |
| SDVT | $1489.8 \pm 507.9$ | $1563.4 \pm 418.1$ | $723.9 \pm 193.8$ | $769.9 \pm 140.0$ |
| Million | $1368.7 \pm 243.3$ | $1248.6 \pm 205.4$ | $563.2 \pm 58.5$ | $562.9 \pm 60.3$ |

the contribution of each component. The maximum success rates, averaged over five random seeds, along with the corresponding return values, are displayed in Table 2. In both benchmarks, the absence of these components led to a reduction in average return. In contrast, incorporating either the coarse-to-fine policy or the holistic-local contrastive representation resulted in an increase in average return and task success rate. Notably, combining the holistic-local contrastive representations (HLR) with the coarse-to-fine policy refinement (CFO) significantly enhanced both success rate and average return across all environments. This improvement can be attributed to our innovative task representations, which structure the latent space at both holistic and local levels, producing more robust task representations. These robust representations enable the generation of effective task-aware policies, thereby enhancing adaptability to new tasks.

Table 2: Ablation study performed on the ML-10 and ML-45 benchmarks, comparing CFOHLR with methods that omit Coarse-to-Fine Policy Refinement (CFO) and Holistic-Local Contrastive Representation (HLR).

| | ML-10 | | ML-45 | |
|---|---|---|---|---|
| **Methods** | **Train** | **Test** | **Train** | **Test** |
| | **Success Rate (%)** | | | |
| Ours | $\mathbf{85.1 \pm 3.5}$ | $\mathbf{85.7 \pm 4.9}$ | $\mathbf{71.4 \pm 4.3}$ | $\mathbf{71.4 \pm 3.5}$ |
| Ours w/o CFO | $82.4 \pm 7.7$ | $45.4 \pm 13.7$ | $66.0 \pm 5.8$ | $33.1 \pm 3.0$ |
| Ours w/o HLR | $82.9 \pm 4.8$ | $49.0 \pm 5.5$ | $69.8 \pm 4.1$ | $33.0 \pm 6.3$ |
| Ours w/o CFO&HLR | $76.4 \pm 22.8$ | $39.4 \pm 13.9$ | $61.4 \pm 11.7$ | $22.6 \pm 7.8$ |
| | **Return** | | | |
| Ours | $\mathbf{3724.8 \pm 219.7}$ | $\mathbf{3733.5 \pm 164.9}$ | $\mathbf{2960.9 \pm 149.0}$ | $\mathbf{2911.7 \pm 105.1}$ |
| Ours w/o CFO | $3651.7 \pm 289.2$ | $1476.4 \pm 224.2$ | $2796.7 \pm 211.6$ | $871.3 \pm 117.3$ |
| Ours w/o HLR | $3717.1 \pm 112.4$ | $1585.3 \pm 289.8$ | $2879.0 \pm 201.0$ | $899.8 \pm 226.3$ |
| Ours w/o CFO&HLR | $3445.0 \pm 859.5$ | $1449.1 \pm 184.2$ | $2666.6 \pm 422.4$ | $725.0 \pm 115.0$ |

## 5 RELATED WORK

**Meta Reinforcement Learning.** Meta-reinforcement learning (Meta-RL) aims to enable agents to quickly adapt to new tasks by leveraging meta-knowledge gained from training on a diverse set

of similar tasks. Meta-RL approaches can be broadly categorized into two types: gradient-based methods (Finn et al., 2017) and context-based methods (Rakelly et al., 2019; Zintgraf et al., 2019). Gradient-based meta-RL methods focus on developing models capable of rapid adaptation to new tasks through a few gradient updates but do not support zero-shot adaptation. In contrast, context-based meta-RL methods comprise a task inference module and a conditional policy module. The task inference module infers task representations from trajectory information, while the conditional policy module guides the agent's action selection based on the environmental state and the inferred task representation. In this paper, we adopt the context-based meta-RL architecture.

**Task-level Adaptation in Meta-RL.**   Most studies in meta-RL focus on narrow task distributions, where different tasks are defined by varying only a few parameters related to the reward function or environment dynamics (Duan et al., 2016; Zintgraf et al., 2019; Rakelly et al., 2019). However, these approaches do not accurately reflect real-world scenarios and limit the agent's ability to adapt to a wide range of tasks, particularly at the task level. Consequently, recent research efforts are directed toward addressing the challenge of task-level adaptation. For example, SDVT (Lee et al., 2023) employs a Gaussian mixture VAE to meta-learn task representations and proposes a virtual training procedure to improve generalization to unseen tasks. Similarly, Million (Bing et al., 2023) integrates transformers with task language instruction to enhance task adaptation capabilities. However, both approaches fail to fully leverage hierarchical characteristic of task-level adaptation in task representations and policy learning, thereby obtain limited gains in adaptation performance.

**Mixture of Expert.**   To enhance performance in completing complex tasks, a promising approach is the use of compositional modules, specifically the mixture of experts (MoE) method (Masoudnia & Ebrahimpour, 2014). The core idea of MoE is to integrate multiple expert models, each specialized in processing a distinct type of input or a specific aspect of a task. These expert models can learn independently and develop specialized capabilities during the training process. For instance, Routing Networks (Rosenbaum et al., 2017) consist of a router and a set of neural network modules; the router selects a module based on the given input and repeats this process iteratively. In contrast, soft modularization (Yang et al., 2020) employs an attention network to generate weights for each module. In this paper, we adopt the mechanism of soft module selection to construct a coarse policy within the coarse-to-fine policy refinement process.

**Contrastive Learning**   To structure the latent space of task representations and enhance their robustness, we employ contrastive learning to improve the task inference process. Previous studies (Li et al., 2020; 2021; Yuan & Lu, 2022; Wang et al., 2023; Gao et al., 2023) have also utilized contrastive learning for this purpose. For instance, FOCAL (Li et al., 2021) introduced a loss function that uses negative-power distance metrics to constrain the task representation space. Similarly, Moss (Wang et al., 2023) employs contrastive learning to differentiate between distinct tasks while clustering similar ones. However, these methods focus exclusively on task instance-level contrastive representation learning, neglecting task category-level contrastive representation learning. This oversight results in a failure to structure the task representation space from a global perspective, thereby reducing the robustness of task representations. To the best of our knowledge, our work is the first to combine the strengths of both instance-wise and category-level contrastive representation methods in meta-RL to achieve robust task representation in task-level adaptation, thereby improving adaptation performance.

## 6   CONCLUSION

In this study, we have introduced a novel method that significantly improves zero-shot performance in task-level adaptation within meta-RL. This enhancement is achieved by integrating a coarse-to-fine policy refinement with a holistic-local contrastive task representation. Specifically, we leverage language instructions to select skill-specific expert modules as a coarse policy. This coarse policy is then refined by a fine policy employing a hypernetwork to generate a task-aware policy based on task representations. To derive robust task representations, we utilize contrastive learning to refine them from both holistic and local perspectives. Experimental results demonstrate that our method substantially boosts learning efficiency and zero-shot adaptation to new tasks, outperforming previous approaches on the Meta-World ML-10 and ML-45 benchmarks.

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

## A  TRAINING APPROACH AND PSEUDOCODE

We utilize Proximal Policy Optimization (PPO) (Schulman et al., 2017) to train our policy network. PPO is an on-policy, actor-critic deep RL algorithm. The optimization objective for the policy is as follows:

$$\mathcal{L}_{\text{policy}}(\theta) = \hat{\mathbb{E}} \left[ \min \left( r_t(\theta) \hat{A}_t, \text{clip} \left( r_t(\theta), 1 - \epsilon, 1 + \epsilon \right) \hat{A}_t \right) \right] \tag{9}$$

Here, $\hat{A}t$ denotes the estimation of the advantage function, and $r_t(\theta)$ represents the probability ratio, defined as $r_t(\theta) = \frac{\pi\theta(a_t|s_t)}{\pi_{\theta_{\text{old}}}(a_t|s_t)}$, where $\pi_\theta$ represents the new policy and $\pi_{\theta_{\text{old}}}$ represents the old policy.

In optimizing the conditional policy module, we utilize the $\mathcal{L}_{\text{policy}}$ loss function. Notably, similar to the approach utilized in VariBAD (Zintgraf et al., 2019), the optimization of the task inference module does not rely on the $\mathcal{L}_{\text{policy}}$ loss function. Instead, we adopt a composite loss function that combines reconstruction and contrastive learning objectives. The specific pseudo-code is shown in Algorithm 1.

---

**Algorithm 1** CFOHLR

---

**Require:** Encoder $q_\phi$ and Decoder $p_\theta$ of VAE; Coarse policy $\pi_\theta$; Fine policy $\pi_\omega$; Weight generator $W_\alpha$ Skill-specific expert models $\text{MoE}_{\theta_i}^{i=0,...,K}$; Hypernetworks $H_\phi$; VAE buffer $\mathcal{D}_{\text{VAE}}$; Policy buffer $\mathcal{D}_{\text{Policy}}$; The number of meta-episodes $n_{\text{meta}}$; The number of rollout episodes per meta-episode $n_{\text{roll}}$; Language instruction $\mathcal{U}$.
  **while** i=0,...,$N_{update}$ **do**
    Sample $K$ training tasks $M_i^{i=0,...,K} \sim M_{\text{train}}$
    **for** timestep t = 0,...,$n_{\text{roll}} * H - 1$ **do**
      **if** $t \bmod H = 0$ **then**
        Reset rollout episode for each task, obtain $S_t = \{s_{t,1}, s_{t,2}, \ldots, s_{t,n}\}$
      **end if**
      **for** j=0,...,K **do**
        Obtain weights $\alpha_{1,j}, \ldots, \alpha_{k,j} = W_\alpha(\mathcal{U}_j)$ for each skill-specific expert module.
        Obtain the output of the coarse policy $\pi_\theta$, denoted as $O_{\text{MOE}} = \sum_{i=0}^{K} \alpha_i \cdot \text{MoE}_{\theta_i}$.
        Leverage $H_\phi$ to generate the network parameters of the fine policy, $\pi_\omega = H_\phi(z_t^j)$.
        Obtain the action $a_{t,j} = \pi_\omega(O_{\text{MOE}})$.
      **end for**
      Finally, obtain actions for each task $A_t = \{a_{t,1}, a_{t,2}, \ldots, a_{t,n}\}$.
      Take an environment step, obtaining $S_{t+1} = \{s_{t+1,1}, s_{t+1,2}, \ldots, s_{t+1,n}\}$ and $R_t = \{r_{t+1,1}, r_{t+1,2}, \ldots, r_{t+1,n}\}$.
      Add the transition $(S_t, A_t, R_{t+1}, S_{t+1})$ to $\mathcal{D}_{\text{VAE}}$ and $\mathcal{D}_{\text{Policy}}$.
      Update task representations $Z_{t+1} = \{z_{t+1,n} = q_\phi(\tau_{:t+1,n})\}_{i=0,...,n}$.
    **end for**
    Update VAE by minimizing $\mathcal{L} = \mathcal{L}_{\text{VAE}} + \mathcal{L}_{\text{contra}}$
    Update policy $\theta, \omega$ and weight generator $\alpha$ by minimizing $L_{\text{actor}} + L_{\text{critic}}$.
  **end while**

---

## B  LIMITATIONS AND FUTURE WORK

Despite the significant progress, our method has limitations that were not addressed in this study. Notably, it is not directly applicable to the cross-entity adaptation problem, which involves generalizing a policy from one robotic entity to another. This limitation affects the overall generalizability of the policy. Future research will focus on tackling the challenge of cross-entity adaptation in a zero-shot manner, thereby enhancing the policy generalization.

## C    IMPLEMENTATION DETAILS

### C.1    REFERENCE IMPLEMENTATIONS

**SDVT, LDM, and VariBAD**    We adapt the SDVT (Lee et al., 2023), LDM (Lee & Chung, 2021), and VariBAD (Zintgraf et al., 2019) algorithms to the Meta-World benchmark. These algorithms are all based on the VariBAD method, which itself is grounded in the Bayesian Adaptive MDP (BAMDP) framework. VariBAD employs a VAE architecture consisting of a recurrent encoder and a dynamics decoder to obtain task representations. LDM introduces a virtual training procedure to VariBAD to address out-of-distribution challenges. Building on LDM, SDVT uses a Gaussian mixture distribution to model the latent space of the VAE. Notably, the virtual training steps of the LDM and SDVT methods are included in the total count of training steps, as these virtual processes necessitate agent interaction with the environment to obtain real states for generating imagined samples. We used open-source code to reproduce the results of the SDVT, LDM, and VariBAD methods, respectively, available at `https://github.com/suyoung-lee/SDVT`, `https://github.com/suyoung-lee/LDM`, and `https://github.com/lmzintgraf/varibad`.

**Million**    Million (Bing et al., 2023) introduces a meta-RL paradigm comprising an instruction phase and a trial phase, integrating transformers with language instruction to improve task adaptation capabilities. We used open-source code to reproduce the results of the Million methods, respectively, available at `https://github.com/yaoxt3/MILLION`.

### C.2    HYPERPARAMETERS

#### C.2.1    SDVT

We used the default hyperparameters from the paper, which are shown in Table 3.

#### C.2.2    LDM AND VARIBAD

We used the default hyperparameters from the paper, which are shown in Table 4.

#### C.2.3    MILLION

We used the default hyperparameters from the paper, which are shown in Table 5.

#### C.2.4    OURS

### C.3    NETWORK ARCHITECTURE

Our method utilizes a context-based architecture, comprising a task inference module and a conditional policy module. For the task inference module, similar to SDVT, we also employ a Gaussian mixture VAE to model the latent space. This module consists of an RNN-based encoder and a prediction decoder. Before being input into the encoder or decoder, all state, action, and reward inputs pass through embedding networks. Regarding the conditional policy module, it includes language-selected, skill-specific expert modules and a hypernetwork-based, task-aware policy. Similarly, before being input into the conditional policy module, all state, action, and reward inputs pass through embedding networks.

### C.4    TASK DESCRIPTIONS

In Table 11, we provide the language instructions for each of the 50 Meta-World tasks.

## D    DETAILED EXPERIMENTAL RESULTS

We adhere to the success criterion established by Meta-World. A timestep is considered successful when the distance between the task-relevant object and the target falls within an acceptable range. Furthermore, an entire rollout episode is deemed successful if the agent achieves success at any timestep during the episode.

Table 3: Hyperparameters used for Garage experiments with SDVT

| Description | ML10 | ML45 |
|---|---|---|
| Meta-Task Hyperparameters | | |
| Meta-batch size | 10 | 10 |
| Tasks sampled per epoch | 10 | 10 |
| General Hyperparameters | | |
| Batch size | 1,000 | 1,000 |
| Path length per roll-out | 1,000 | 1,000 |
| Discount factor | 0.99 | 0.99 |
| Algorithm-Specific Hyperparameters | | |
| Policy hidden sizes | $(256, 256)$ | $(256, 256)$ |
| Activation function | tanh | tanh |
| Policy learning rate | $7 \times 10^{-4}$ | $7 \times 10^{-4}$ |
| PPO epochs num | 5 | 5 |
| VAE learning rate | $1 \times 10^{-3}$ | $1 \times 10^{-3}$ |
| Latent dimension | 5 | 5 |
| PPO num minibatches | 10 | 10 |
| PPO clip param | 0.1 | 0.1 |
| Policy num steps | 5 | 5 |
| Size of VAE buffer | 1,000 | 1,000 |
| KL weight | 0.1 | 0.1 |
| VAE mixture num | 10 | 10 |
| Gaussian loss coefficient | 1.0 | 1.0 |
| Action embedding size | 16 | 16 |
| State embedding size | 32 | 32 |
| Reward embedding size | 16 | 16 |
| Virtual ratio increment | 0.05 | 0.05 |
| Number of virtual skills | 3 | 3 |
| RL loss through encoder | False | False |
| VAE loss coefficient | 1.0 | 1.0 |

## D.1 PERFORMANCE ON INDIVIDUAL TASKS

### D.1.1 ML-10

### D.1.2 ML-45

## D.2 LEARNING CURVES

In Figure 5, we present the mean and standard deviation of returns and success rates across five random seeds.

## E ADDITIONAL RESULTS

### E.1 VISUALIZATIONS

To demonstrate the quality of the learned task representations, we employed t-SNE Van der Maaten & Hinton (2008) to map the task representation vectors into a 2D space, enabling the visualization of these representations. For each testing task, 150 transitions from the meta-testing phase were sampled to visualize the task representations. As depicted in Figure 6, our method effectively distinguishes task representations from different categories, with additional separation observed among tasks within the same category.

Table 4: The hyperparameters used in experiments with LDM and VariBAD are consistent across both models in the general and policy categories of SDVT, as outlined in Table 3. The only difference lies in the modeling of the latent space: SDVT utilizes a Gaussian mixture model, while both LDM and VariBAD employ a Gaussian model.

| Description | ML10 | ML45 |
|---|---|---|
| Meta-Task Hyperparameters | | |
| Meta-batch size | 10 | 10 |
| Tasks sampled per epoch | 10 | 10 |
| General Hyperparameters | | |
| Batch size | 1,000 | 1,000 |
| Path length per roll-out | 1,000 | 1,000 |
| Discount factor | 0.99 | 0.99 |
| Algorithm-Specific Hyperparameters | | |
| VAE learning rate | $1 \times 10^{-3}$ | $1 \times 10^{-3}$ |
| Latent dimension | 5 | 5 |
| Size of VAE buffer | 1,000 | 1,000 |
| KL weight | 0.1 | 0.1 |
| Gaussian loss coefficient | 1.0 | 1.0 |
| VAE loss coefficient | 1.0 | 1.0 |

Table 5: Hyperparameters used in experiments with Million.

| Description | ML10 | ML45 |
|---|---|---|
| Meta-Task Hyperparameters | | |
| Meta-batch size | 10 | 10 |
| Tasks sampled per epoch | 10 | 10 |
| General Hyperparameters | | |
| Batch Timesteps | 1,000 | 1,000 |
| Action repeat | 1,000 | 1,000 |
| Demonstration action repeat | 1,000 | 1,000 |
| Max trials per episode | 750 | 750 |
| Discount factor | 0.99 | 0.99 |
| Algorithm-Specific Hyperparameters | | |
| Learning rate | $1e-4$ | $1e-4$ |
| GAE lambda | 0.97 | 0.97 |
| Epsilon eta | $1 \times 10^{-2}$ | $1 \times 10^{-2}$ |
| Epsilon alpha | $1 \times 10^{-2}$ | $1 \times 10^{-2}$ |
| Epsilon alpha mu | 0.0075 | 0.0075 |
| Epsilon alpha sigma | $1e-5$ | $1e-5$ |

Table 6: Hyperparameters used in experiments with Ours.

| Description | ML10 | ML45 |
|---|---|---|
| **Meta-Task Hyperparameters** | | |
| Meta-batch size | 10 | 10 |
| Tasks sampled per epoch | 10 | 10 |
| **General Hyperparameters** | | |
| Batch size | 1,000 | 1,000 |
| Path length per roll-out | 1000 | |
| Discount factor | 0.99 | |
| **Algorithm-Specific Hyperparameters** | | |
| Policy hidden sizes | $(256, 256)$ | $(256, 256)$ |
| Activation function | tanh | tanh |
| Policy learning rate | $7 \times 10^{-4}$ | $7 \times 10^{-4}$ |
| PPO epochs num | 5 | 5 |
| VAE learning rate | $1 \times 10^{-3}$ | $1 \times 10^{-3}$ |
| Latent dimension | 5 | 5 |
| PPO num minibatches | 10 | 10 |
| PPO clip param | 0.1 | 0.1 |
| Policy num steps | 5 | 5 |
| RL loss through encoder | False | False |
| Action embedding size | 16 | 16 |
| State embedding size | 32 | 32 |
| Reward embedding size | 16 | 16 |
| Size of VAE buffer | 1,000 | 1,000 |
| KL weight | 0.1 | 0.1 |
| VAE mixture num | 10 | 10 |
| Gaussian loss coefficient | 1.0 | 1.0 |
| VAE loss coefficient | 1.0 | 1.0 |
| Decode reward | True | True |
| Decode state | True | True |
| Weight of holistic contrastive | 0.01 | 0.01 |
| Weight of local contrastive | 0.01 | 0.01 |

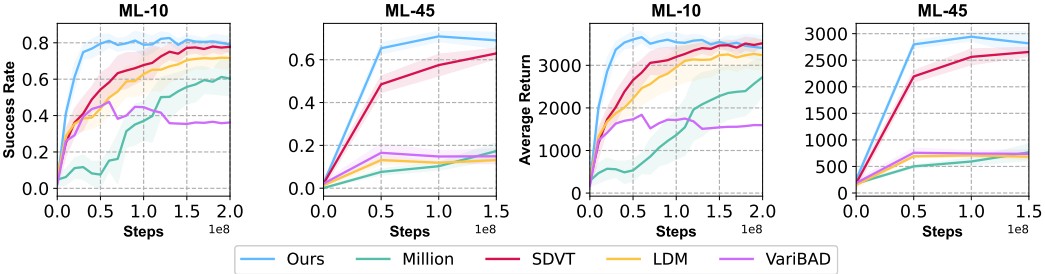

Figure 5: **Learning Curves on ML-10 and ML-45.** The maximum success rates and corresponding returns of our methods, along with baseline comparisons, are presented. The plot shows the mean and standard deviation of returns across five random seeds.

Table 7: ML-10 of Meta-World success rate (%). We present the final success rates of our method and baseline approaches on both the training and test tasks of the Meta-World ML-10 benchmark. All results are reported as the mean success rate $\pm 95\%$ confidence interval of five seeds.

| Index. Task | Ours | w/o C2F | w/o HLC | SDVT | Million | LDM | VariBAD |
|---|---|---|---|---|---|---|---|
| 1. Reach | **85.2±3.6** | 53.2±4.6 | 44.0±7.3 | 53.6±14.4 | 10.4±11.3 | 50.4±7.9 | 78.0±5.7 |
| 2. Push | **86.0±3.0** | 70.4±9.8 | 74.8±3.0 | 74.0±3.8 | 44.5±17.8 | 31.2±11.2 | 2.4±2.0 |
| 3. Pick-place | **85.2±4.5** | 66.0±6.9 | 66.0±2.9 | 53.2±6.0 | 48.1±29.8 | 37.2±20.0 | 0.8±0.7 |
| 4. Door-open | 81.2±1.9 | 99.6±0.6 | 97.2±3.9 | **100.0±0.0** | 81.1±25.1 | 99.6±0.6 | 74.8±25.4 |
| 5. Drawer-close | 86.8±1.4 | **100.0±0.0** | **100.0±0.0** | **100.0±0.0** | 56.1±32.1 | **100.0±0.0** | **100.0±0.0** |
| 6. Button-press | 84.4±3.1 | **100.0±0.0** | **100.0±0.0** | 99.6±0.6 | 80.0±27.7 | 98.4±1.0 | 88.4±4.1 |
| 7. Peg-insert-side | **88.0±2.5** | 48.8±18.6 | 62.0±13.1 | 52.0±5.0 | 21.5±18.3 | 26.4±15.3 | 0.0±0.0 |
| 8. Window-open | 86.4±4.4 | 99.6±0.6 | **100.0±0.0** | **100.0±0.0** | 80.0±27.7 | 99.6±0.6 | 96.8±1.1 |
| 9. Sweep | 89.6±2.7 | **93.2±2.1** | 93.2±3.5 | 89.2±3.8 | 77.6±24.0 | 92.4±3.3 | 0.0±0.0 |
| 10. Basketball | 82.8±4.5 | **93.6±4.0** | 92.0±5.6 | 72.8±15.5 | 36.1±31.0 | 89.2±4.0 | 0.0±0.0 |
| **Train mean** | **85.6±3.9** | 82.4±7.7 | 82.9±4.8 | 79.4±7.1 | 53.5±40.0 | 72.4±11.4 | 44.1±7.9 |
| 11. Drawer-open | 86.4±3.4 | 78.0±24.1 | 80.0±12.6 | 72.8±28.5 | 0.0±0.0 | **92.8±8.1** | 51.6±21.1 |
| 12. Door-close | 87.2±3.8 | 74.4±25.7 | 81.6±15.7 | 76.4±19.7 | **97.5±2.8** | 26.0±37.7 | 71.6±22.4 |
| 13. Shelf-place | **82.4±5.5** | 1.6±3.1 | 0.4±0.8 | 0.0±0.0 | 0.3±0.5 | 0.4±0.8 | 0.0±0.0 |
| 14. Sweep-into | **90.0±5.1** | 67.6±34.0 | 82.8±8.8 | 64.8±12.2 | 17.5±5.0 | 57.6±31.9 | 4.8±3.2 |
| 15. Lever-pull | **82.4±3.4** | 5.2±7.4 | 0.4±0.8 | 3.2±3.2 | 13.7±26.9 | 0.4±0.8 | 0.4±0.8 |
| **Test mean** | **85.7±4.9** | 45.4±13.7 | 49.0±5.5 | 43.4±9.4 | 25.8±9.1 | 35.4±17.1 | 25.7±7.5 |

Table 8: ML-10 of Meta-World returns. We present the performance metrics of our method and baseline approaches on both the training and test tasks of the Meta-World ML-10 benchmark. All results are reported as the mean return $\pm 95\%$ confidence interval of five seeds.

| Index. Task | Ours | w/o C2F | w/o HLC | SDVT | Million | LDM | VariBAD |
|---|---|---|---|---|---|---|---|
| 1. Reach | 3704±149 | **3778±128** | 3520±141 | 3763±296 | 2324±447 | 3668±285 | 4054±138 |
| 2. Push | **3769±135** | 3338±342 | 4094±90 | 3675±272 | 2225±750 | 1795±812 | 63±28 |
| 3. Pick-place | **3742±127** | 2089±254 | 2420±116 | 1712±125 | 1678±803 | 1258±709 | 7±1 |
| 4. Door-open | 3740±91 | 4503±51 | 4313±78 | 4439±26 | 2790±980 | **4442±47** | 2978±470 |
| 5. Drawer-close | 3708±75 | **4857±7** | 4811±30 | 4852±10 | 2505±1302 | 4809±67 | 4637±77 |
| 6. Button-press | 3622±144 | 3489±93 | 3250±108 | **3372±60** | 1337±734 | 3226±60 | 2028±155 |
| 7. Peg-insert-side | **3703±113** | 2359±678 | 2827±421 | 2443±266 | 1179±697 | 1364±773 | 9±1 |
| 8. Window-open | 3787±157 | **4479±49** | 4398±49 | 4476±51 | 2331±942 | 4384±61 | 3692±202 |
| 9. Sweep | 3786±147 | **4093±85** | 3963±100 | 3801±208 | 2849±932 | 3997±189 | 92±28 |
| 10. Basketball | **3705±185** | 3532±196 | 3576±202 | 2937±618 | 1624±774 | 3433±196 | 9±2 |
| **Train mean** | **3727±221** | 3652±289 | 3717±112 | 3547±203 | 2084±1365 | 3238±613 | 1757±178 |
| 11. Drawer-open | **3796±112** | 2477±393 | 2477±190 | 2660±396 | 1876±384 | 2697±475 | 2036±329 |
| 12. Door-close | **3740±119** | 2489±566 | 2887±769 | 3087±944 | 3302±415 | 1272±1538 | 2113±558 |
| 13. Shelf-place | **3746±157** | 492±246 | 607±115 | 341±99 | 141±204 | 309±272 | 0±0 |
| 14. Sweep-into | **3751±128** | 1619±786 | 1705±434 | 1444±564 | 716±264 | 1200±793 | 172±96 |
| 15. Lever-pull | **3634±197** | 305±44 | 251±29 | 285±60 | 208±44 | 278±59 | 324±41 |
| **Test mean** | **3734±165** | 1476±224 | 1585±290 | 1563±418 | 1249±205 | 1151±692 | 929±208 |

Table 9: ML-45 of Meta-World success rate (%). We present the final success rates of our method and baseline approaches on both the training and test tasks of the Meta-World ML-45 benchmark. All results are reported as the mean success rate $\pm 95\%$ confidence interval of five seeds.

| Index. Task | Ours | w/o C2F | w/o HLC | SDVT | Million | LDM | VariBAD |
|---|---|---|---|---|---|---|---|
| 1. Assembly | **68.0**±**2.0** | 0.5±0.3 | 0.5±0.3 | 0.5±0.3 | 0.0±0.0 | 0.0±0.0 | 0.0±0.0 |
| 2. Basketball | **67.5**±**1.3** | 22.0±5.7 | 21.5±1.7 | 38.0±5.2 | 0.0±0.0 | 0.0±0.0 | 0.0±0.0 |
| 3. Button-press-topdown | 65.0±3.2 | **99.0**±**0.6** | 98.5±0.3 | 98.0±0.7 | 0.0±0.0 | 9.5±2.8 | 27.5±6.7 |
| 4. Button-press-topdown-wall | 72.0±2.3 | 96.0±2.3 | 98.5±0.6 | **99.0**±**0.3** | 0.0±0.0 | 9.5±2.7 | 17.0±4.6 |
| 5. Button-press-wall | 75.5±3.0 | 94.0±2.1 | 95.0±1.8 | **100.0**±**0.0** | 43.2±10.8 | 30.0±3.9 | 27.0±6.1 |
| 6. Button-press-wall | 71.0±2.4 | **88.0**±**3.1** | 76.0±4.8 | 80.5±4.2 | 49.2±6.0 | 41.0±9.7 | 32.0±4.8 |
| 7. Coffee-button | 65.0±2.8 | **100.0**±**0.0** | **100.0**±**0.0** | 98.5±0.9 | 42.3±14.7 | 55.5±11.0 | 33.5±8.7 |
| 8. Coffee-pull | **73.5**±**2.9** | 62.5±1.3 | 70.0±3.6 | 37.0±5.8 | 0.2±0.1 | 1.0±0.3 | 0.0±0.0 |
| 9. Coffee-push | **76.5**±**1.5** | 57.5±10.7 | 69.5±5.5 | 37.5±4.9 | 30.8±9.7 | 11.5±3.3 | 7.5±2.1 |
| 10. Dial-turn | 73.5±2.6 | 73.0±6.1 | 75.0±4.9 | **77.5**±**2.6** | 62.2±2.4 | 8.5±1.7 | 26.0±6.4 |
| 11. Disassemble | 74.0±3.2 | 69.5±13.6 | **79.0**±**5.3** | 78.0±4.3 | 0.0±0.0 | 0.5±0.3 | 4.5±2.6 |
| 12. Door-close | 72.5±1.3 | **100.0**±**0.0** | 99.5±0.3 | 100.0±0.0 | 51.0±11.9 | 98.5±0.9 | 65.0±13.8 |
| 13. Door-open | 74.5±2.8 | 92.0±3.3 | 94.0±2.7 | **95.5**±**0.9** | 0.0±0.0 | 0.0±0.0 | 0.0±0.0 |
| 14. Drawer-close | 70.5±1.7 | 96.0±1.4 | **100.0**±**0.0** | 99.5±0.3 | **100.0**±**0.0** | 99.0±0.6 | **100.0**±**0.0** |
| 15. Drawer-open | 68.0±2.4 | **99.5**±**0.3** | 95.0±1.5 | 98.5±0.6 | 0.0±0.0 | 15.0±6.1 | 10.0±2.4 |
| 16. Faucet-open | 71.0±2.7 | 98.0±0.5 | 96.0±2.0 | **98.5**±**0.9** | 11.2±3.8 | 30.5±1.5 | 54.0±9.7 |
| 17. Faucet-close | 77.0±1.0 | 98.5±0.9 | 87.5±3.8 | **99.5**±**0.3** | 42.0±13.5 | 22.0±4.4 | 28.5±7.1 |
| 18. Hammer | **75.5**±**1.5** | 8.5±5.0 | 33.0±11.1 | 0.0±0.0 | 12.2±7.1 | 1.5±0.9 | 2.0±0.8 |
| 19. Handle-press-side | 69.0±1.4 | **100.0**±**0.0** | 99.0±0.6 | **100.0**±**0.0** | 14.2±4.9 | 6.0±2.2 | 38.5±8.5 |
| 20. Handle-press | 70.0±1.3 | **100.0**±**0.0** | 99.5±0.3 | **100.0**±**0.0** | 65.5±3.2 | 45.5±2.9 | 57.0±2.5 |
| 21. Handle-pull-side | 72.5±3.0 | 96.0±2.0 | **98.0**±**1.2** | 89.5±3.6 | 17.7±4.6 | 0.0±0.0 | 1.5±0.6 |
| 22. Handle-pull | 70.5±1.9 | 76.5±13.3 | **99.5**±**0.3** | 63.5±12.2 | 27.0±10.1 | 1.5±0.3 | 1.0±0.6 |
| 23. Lever-pull | **76.5**±**1.2** | 55.5±9.6 | 9.5±5.6 | 52.0±10.4 | 0.0±0.0 | 0.0±0.0 | 1.5±0.3 |
| 24. Peg-insert-side | **72.0**±**1.7** | 9.0±1.1 | 3.5±2.5 | 3.5±0.6 | 0.0±0.0 | 0.0±0.0 | 0.0±0.0 |
| 25. Pick-place-wall | **72.0**±**1.1** | 61.5±4.2 | 59.5±6.4 | 46.0±3.5 | 13.5±5.8 | 0.0±0.0 | 0.5±0.3 |
| 26. Pick-out-of-hole | **67.0**±**3.6** | 39.0±9.9 | 52.0±5.0 | 53.5±5.7 | 0.0±0.0 | 0.0±0.0 | 0.0±0.0 |
| 27. Push | **79.5**±**3.0** | 38.0±3.0 | 48.0±2.2 | 27.5±3.2 | 7.7±2.9 | 42.0±3.8 | 46.0±5.8 |
| 28. Push-back | 66.5±1.9 | 69.0±5.7 | **79.5**±**3.1** | 64.0±4.2 | 0.0±0.0 | 0.0±0.0 | 1.0±0.6 |
| 29. Push | **74.5**±**2.6** | 43.0±4.2 | 68.0±4.9 | 38.5±7.5 | 8.3±2.4 | 4.5±1.0 | 3.0±1.0 |
| 30. Pick-place | **67.5**±**1.2** | 50.5±2.4 | 58.5±2.8 | 47.5±3.6 | 10.3±4.3 | 0.0±0.0 | 1.0±0.3 |
| 31. Plate-slide-side | **69.0**±**2.8** | 47.5±4.8 | 48.0±3.7 | 67.0±3.8 | 36.7±7.0 | 0.0±0.0 | 0.5±0.3 |
| 32. Plate-slide-side | 71.0±3.2 | **95.0**±**1.0** | 91.5±2.2 | 92.5±2.3 | 0.0±0.0 | 0.5±0.3 | 13.5±7.9 |
| 33. Plate-slide back | 67.5±1.5 | 96.5±0.6 | **98.5**±**0.9** | 91.5±1.5 | 0.0±0.0 | 1.5±0.6 | 4.5±1.5 |
| 34. Plate-slide-back-side | 75.5±2.4 | 80.5±5.1 | **89.0**±**2.5** | 83.5±2.7 | 0.0±0.0 | 0.5±0.3 | 6.0±2.0 |
| 35. Peg-unplug-side | 69.5±1.7 | 66.5±3.7 | **76.0**±**4.7** | 53.5±3.8 | 5.3±1.3 | 6.5±2.8 | 4.5±1.5 |
| 36. Soccer | **73.0**±**1.7** | 21.5±4.8 | 18.0±1.2 | 26.0±2.4 | 10.5±2.1 | 9.0±2.2 | 8.0±1.3 |
| 37. Stick-push | **72.0**±**1.4** | 0.0±0.0 | 0.0±0.0 | 0.0±0.0 | 0.0±0.0 | 0.0±0.0 | 0.0±0.0 |
| 38. Stick-pull | **73.5**±**1.9** | 0.0±0.0 | 0.5±0.3 | 0.0±0.0 | 0.0±0.0 | 0.0±0.0 | 0.0±0.0 |
| 39. Push-wall | 72.0±2.4 | 54.5±6.4 | 88.0±1.1 | 54.0±10.0 | 7.8±3.1 | 0.5±0.3 | 1.0±0.6 |
| 40. Reach-wall | 75.0±0.6 | 40.5±6.9 | 60.0±4.1 | 26.5±6.1 | 13.3±5.3 | 49.5±5.6 | **75.0**±**5.2** |
| 41. Shelf-place | **74.5**±**2.5** | 6.0±2.8 | 1.0±0.6 | 1.0±0.6 | 0.8±0.5 | 0.0±0.0 | 0.0±0.0 |
| 42. Sweep-into | 66.0±1.4 | 94.5±1.2 | **95.5**±**1.3** | 81.5±8.1 | 25.0±4.7 | 9.0±1.9 | 11.0±2.6 |
| 43. Sweep | 73.0±1.1 | 74.5±2.9 | **83.0**±**3.6** | 36.5±12.3 | 0.0±0.0 | 0.0±0.0 | 0.0±0.0 |
| 44. Window-open | 64.5±0.7 | **99.0**±**0.3** | 98.5±0.9 | 100.0±0.0 | 60.8±7.9 | 13.5±3.6 | 33.5±6.2 |
| 45. Window-close | 67.0±2.7 | **100.0**±**0.0** | 99.5±0.3 | 99.5±0.3 | 11.8±4.0 | 17.0±3.2 | 29.5±8.0 |
| **Train mean** | **71.4**±**4.3** | 66.0±5.8 | 69.8±4.1 | 63.0±5.5 | 17.3±8.9 | 14.2±2.6 | 17.2±4.2 |
| 46. Bin-picking | **76.0**±**3.2** | 1.0±1.0 | 1.5±0.9 | 3.5±2.6 | 0.2±0.3 | 0.0±0.0 | 0.0±0.0 |
| 47. Box-close | **70.0**±**4.3** | 1.0±1.0 | 6.5±3.9 | 0.5±0.9 | 1.7±2.9 | 0.5±0.9 | 0.5±0.9 |
| 48. Hand-insert | **69.5**±**2.6** | 0.5±0.9 | 2.5±2.6 | 3.5±3.6 | 7.5±7.7 | 3.0±3.4 | 3.0±2.3 |
| 49. Door-lock | 73.0±10.0 | **82.0**±**3.8** | 70.0±17.2 | 61.5±13.4 | 34.3±4.8 | 41.5±10.1 | 34.5±12.6 |
| 50. Door-unlock | 68.5±3.6 | 81.0±7.1 | **84.5**±**9.7** | 66.0±15.5 | 14.2±24.8 | 21.0±10.0 | 37.0±15.0 |
| **Test mean** | **71.4**±**3.5** | 33.1±3.0 | 33.0±6.3 | 27.0±8.9 | 11.6±7.1 | 13.2±6.0 | 15.0±6.5 |

Table 10: ML-45 of Meta-World returns. We present the final returns of our method and baseline approaches on both the training and test tasks of the Meta-World ML-45 benchmark. All results are reported as the mean return $\pm 95\%$ confidence interval of five seeds.

| Index. Task | Ours | w/o C2F | w/o HLC | SDVT | Million | LDM | VariBAD |
|---|---|---|---|---|---|---|---|
| 1. Assembly | **2898**±97 | 2847±27 | 2529±88 | 2590±61 | 329±53 | 154±27 | 101±13 |
| 2. Basketball | **2885**±101 | 1417±116 | 1444±107 | 1569±169 | 281±86 | 5±0 | 11±2 |
| 3. Button-press-topdown | 2693±116 | 3582±100 | **3712**±27 | 3586±50 | 884±128 | 988±104 | 1182±84 |
| 4. Button-press-topdown-wall | 2950±69 | 3541±120 | **3686**±52 | 3594±62 | 868±132 | 977±99 | 1209±81 |
| 5. Button-press-wall | 3186±28 | 3143±111 | 3072±48 | **3193**±101 | 553±102 | 667±68 | 615±96 |
| 6. Button-press-wall | 3118±73 | **3344**±108 | 3170±64 | 3275±44 | 495±108 | 711±156 | 633±87 |
| 7. Coffee-button | 2746±53 | **3465**±71 | 2731±336 | 3309±96 | 684±249 | 204±13 | 211±25 |
| 8. Coffee-pull | **3123**±79 | 1209±45 | 1385±128 | 877±118 | 58±4 | 40±2 | 40±4 |
| 9. Coffee-push | **3197**±39 | 1175±203 | 1463±193 | 729±55 | 228±62 | 41±4 | 65±16 |
| 10. Dial-turn | 2861±70 | **3711**±134 | 3396±108 | 3607±197 | 670±30 | 942±98 | 803±76 |
| 11. Disassemble | **2990**±85 | 2823±431 | 2811±291 | 2878±254 | 124±20 | 156±11 | 130±10 |
| 12. Door-close | 2975±93 | 4310±44 | 4328±63 | **4481**±19 | 1138±161 | 4359±27 | 2661±580 |
| 13. Door-open | 3080±50 | 4004±116 | 3948±76 | **4010**±109 | 636±61 | 624±76 | 607±62 |
| 14. Drawer-close | 2936±35 | 4443±125 | 4730±28 | **4748**±22 | 3625±462 | 4375±92 | 4482±122 |
| 15. Drawer-open | 2855±112 | 4638±6 | 4065±99 | **4391**±14 | 1746±95 | 1284±71 | 1356±127 |
| 16. Faucet-open | 2945±94 | 4608±9 | 4276±158 | **4636**±17 | 1669±47 | 2212±73 | 2584±299 |
| 17. Faucet-close | 3074±23 | **4594**±31 | 3997±173 | 4533±42 | 2349±214 | 2193±129 | 2174±185 |
| 18. Hammer | **3013**±68 | 516±28 | 1299±301 | 468±5 | 563±56 | 394±27 | 397±25 |
| 19. Handle-press-side | 2949±27 | 4689±52 | 4707±29 | **4783**±8 | 480±68 | 489±71 | 1377±270 |
| 20. Handle-press | 2941±53 | **4648**±66 | 4601±48 | 4579±64 | 2063±73 | 1791±100 | 2126±116 |
| 21. Handle-pull-side | 2959±56 | **3647**±143 | 3060±275 | 3340±240 | 165±64 | 19±2 | 24±2 |
| 22. Handle-pull | 3000±83 | 3482±342 | **4019**±59 | 2996±260 | 996±392 | 40±8 | 87±23 |
| 23. Lever-pull | **2999**±46 | 762±79 | 381±33 | 878±89 | 276±13 | 240±10 | 232±11 |
| 24. Peg-insert-side | **2978**±42 | 1307±81 | 1616±128 | 1238±50 | 192±36 | 11±0 | 10±1 |
| 25. Pick-place-wall | **3102**±60 | 2542±157 | 2728±136 | 1812±98 | 491±181 | 0±0 | 2±0 |
| 26. Pick-out-of-hole | **2808**±91 | 781±183 | 1206±142 | 922±122 | 23±5 | 10±1 | 13±1 |
| 27. Push | 3157±49 | 2839±131 | **3313**±108 | 2823±142 | 1555±125 | 3105±126 | 3193±89 |
| 28. Push-back | **2897**±111 | 1284±88 | 1799±66 | 881±203 | 16±2 | 7±1 | 5±0 |
| 29. Push | 3094±115 | 2428±123 | **3421**±56 | 2247±92 | 651±156 | 55±6 | 60±8 |
| 30. Pick-place | **2921**±54 | 1632±76 | 2165±83 | 1449±141 | 291±115 | 8±0 | 10±1 |
| 31. Plate-slide-side | 2942±63 | 2516±138 | 2207±77 | **3221**±98 | 1929±195 | 359±18 | 546±40 |
| 32. Plate-slide-side | 3022±114 | 3250±84 | 2873±85 | **3784**±178 | 818±54 | 191±31 | 662±139 |
| 33. Plate-slide back | 2764±46 | **4235**±28 | 4109±20 | 4165±52 | 1021±52 | 556±15 | 561±67 |
| 34. Plate-slide-back-side | 3021±29 | 3776±179 | **4173**±24 | 4124±58 | 631±142 | 183±35 | 728±134 |
| 35. Peg-unplug-side | **2935**±36 | 1390±217 | 1984±187 | 890±103 | 43±9 | 33±3 | 27±1 |
| 36. Soccer | **2974**±67 | 1056±25 | 1079±41 | 1052±65 | 515±152 | 278±30 | 321±16 |
| 37. Stick-push | **3074**±21 | 612±206 | 289±166 | 26±8 | 125±28 | 11±1 | 14±2 |
| 38. Stick-pull | **2980**±88 | 289±92 | 82±43 | 16±3 | 129±37 | 11±1 | 12±1 |
| 39. Push-wall | 2925±74 | 2224±97 | **3586**±50 | 2548±308 | 854±218 | 33±1 | 48±10 |
| 40. Reach-wall | 3063±41 | 2747±263 | **3447**±180 | 2330±273 | 1602±167 | 2906±263 | 3509±46 |
| 41. Shelf-place | **3041**±76 | 878±70 | 838±105 | 899±36 | 69±35 | 0±0 | 1±0 |
| 42. Sweep-into | 2756±79 | 3962±140 | **4061**±91 | 3095±422 | 879±157 | 238±43 | 207±45 |
| 43. Sweep | 2902±59 | 2976±139 | **3251**±110 | 1640±421 | 325±62 | 65±8 | 83±12 |
| 44. Window-open | 2684±13 | 4142±51 | 4125±87 | **4225**±31 | 710±62 | 471±30 | 795±116 |
| 45. Window-close | 2828±112 | 4392±39 | 4397±28 | **4406**±49 | 710±128 | 928±36 | 1152±96 |
| **Train mean** | **2961**±149 | 2797±212 | 2879±201 | 2685±154 | 766±352 | 719±63 | 779±171 |
| 46. Bin-picking | **2993**±138 | 84±48 | 133±38 | 104±35 | 20±9 | 20±6 | 15±1 |
| 47. Box-close | **2844**±133 | 159±28 | 255±78 | 145±19 | 231±72 | 248±20 | 209±34 |
| 48. Hand-insert | **2708**±238 | 221±41 | 258±67 | 273±111 | 118±89 | 82±63 | 124±102 |
| 49. Door-lock | **2960**±335 | 2048±106 | 1901±747 | 1565±227 | 1659±184 | 1692±338 | 1589±374 |
| 50. Door-unlock | **2962**±51 | 1844±222 | 1952±206 | 1763±264 | 787±217 | 944±167 | 1219±339 |
| **Test mean** | **2912**±105 | 871±117 | 900±226 | 770±140 | 563±60 | 597±154 | 631±203 |

Table 11: A list of all of the Meta-World tasks and a description of each task.

| Task | Language instructions |
| --- | --- |
| assembly | pick up a nut and place it onto a peg |
| basketball | pick the basketball and place at the goal point |
| button-press-topdown | push the button down to the goal point |
| button-press-topdown-wall | bypass a wall and press a button from the top |
| button-press | press a button |
| button-press-wall | bypass a wall and press a button |
| coffee-button | push a button on the coffee machine |
| coffee-pull | place cup away |
| coffee-push | push cup to the goal point |
| dial-turn | rotate a dial 180 degrees |
| disassemble | pick a nut out of a peg |
| door-close | push the door to the goal point |
| door-open | pull the door to the goal point |
| drawer-close | push the drawer to the goal point |
| drawer-open | pull the drawer to the goal point |
| faucet-open | rotate the faucet counter-clockwise |
| faucet-close | rotate the faucet clockwise |
| hammer | push to the goal point with hammer |
| handle-press-side | press a handle down sideways |
| handle-press | press a handle down |
| handle-pull-side | pull a handle up sideways |
| handle-pull | pull a handle up |
| lever-pull | pull the lever to the goal point |
| peg-insert-side | insert the peg to the goal point |
| pick-place-wall | pick a puck, bypass a wall and place the puck |
| pick-out-of-hole | pick up a puck from a hole |
| reach | reach the goal point |
| push-back | push the puck back to the goal point |
| push | push the puck to the goal point |
| pick-place | pick the puck and place at the goal point |
| plate-slide | push the plate to the goal point |
| plate-slide-side | push the plate left to the goal point |
| plate-slide-back | push the plate back to the goal point |
| plate-slide-back-side | push the plate right to the goal point |
| peg-unplug-side | pull a peg sideways to the goal point |
| soccer | push a ball to the goal point |
| stick-push | grasp a stick and push a box using the stick |
| stick-pull | grasp a stick and pull a box with the stick |
| push-wall | bypass a wall and push a puck to a goal |
| reach-wall | bypass a wall and reach a goal |
| shelf-place | pick the puck and place on shelf at the goal point |
| sweep-into | sweep the puck into the box |
| sweep | sweep the puck off the table |
| window-open | push the window to the goal point |
| window-close | push the window to the goal point |
| bin-picking | grasp the puck from one bin and place it into another bin |
| box-close | grasp the cover and close the box with it |
| hand-insert | insert the gripper into a hole |
| door-lock | rotate the lock clockwise |
| door-unlock | rotate the lock counter-clockwise |

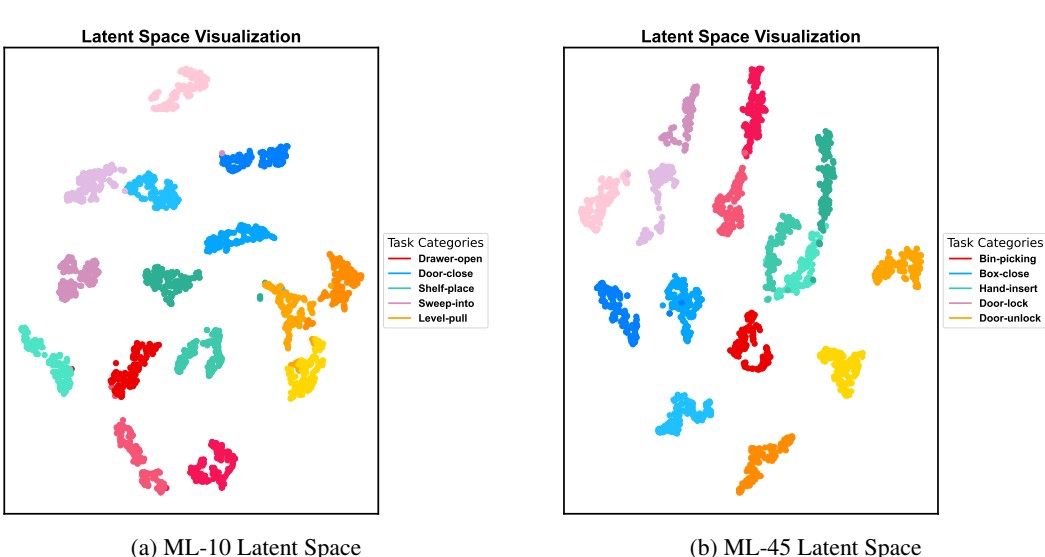

(a) ML-10 Latent Space          (b) ML-45 Latent Space

Figure 6: **Latent Space Visualization**. The t-SNE visualization of the learned task representation space for the ML-10 testing tasks is presented. We sampled three tasks from each task category of the test tasks, with each color scheme representing a different task category. Each point in the visualization corresponds to a task representation vector extracted from transitions and is color-coded according to the task properties.

