# OpenReview forum: "Zero-Shot Task-Level Adaptation via Coarse-to-Fine Policy Refinement and Holistic-Local Contrastive Representations"
_ICLR.cc/2025/Conference — ICLR 2025 Conference Withdrawn Submission_

### Official Review · Reviewer_ZS5D · 2024-10-29

**Soundness:** 2
**Presentation:** 3
**Contribution:** 2
**Rating:** 5
**Confidence:** 3

**Summary:**

This paper introduces a novel meta-RL approach to improve zero-shot task adaptation.

By integrating Coarse-to-Fine Policy Refinement (CFO) and Holistic-Local Contrastive Task Representation (HLR), the method enables efficient adaptation to new tasks. CFO leverages natural language descriptions for skill module selection and uses a hypernetwork for fine-grained policy adjustment, while HLR employs contrastive learning to create robust task representations across categories and instances.

Experimental results on Meta-World benchmarks (ML-10, ML-45) show that this approach significantly outperforms baselines in success rates and returns, demonstrating its effectiveness.

**Strengths:**

1. This paper addresses the problem of task-level zero-shot adaptation in meta-RL, which is a very important topic.
2. The idea of coarse-to-fine policy refinement is novel. It can first capture the general skills required for the task, and then, through a hypernetwork, generate a fine-grained, task-aware policy that adapts to the specific attributes of the task.
3. The paper is clear and well-written.
4. The two insights the authors point out, "effective task-level adaptation requires an agent to have a general understanding of task forms and to select appropriate skills accordingly" and "developing an effective task-aware policy depends on accurately capturing task attributes through robust task representations", has a potential to motivate new research.

**Weaknesses:**

1. It is unclear why this method can significantly improve zero-shot adaptation, since it lacks some necessary analysis. See the questions in the next section.
2. The experiments need improvement. The benchmark is limited to Meta-World, and if my understanding is correct, it was only tested in two scenarios where both the training and test sets are fixed. This raises concerns about the potential for overfitting.
3. Section D.1 in the appendix is incomplete.
4. The corresponding code for the experiment has not been provided. Since the experimental results show significant improvements over previous work, providing the code would further validate its effectiveness.

**Questions:**

1. One of the key focuses of this paper is "Holistic-Local Contrastive Representation." Could you visualize and analyze whether their representations are distributed as expected, highlighting the differences compared to representations from previous methods?
2. For coarse-to-fine refinement part, analyzing the activation of the MoE module is crucial. For instance, demonstrating which experts are activated by different instructions could help reveal underlying patterns and provide deeper insights.
3. The ablation section needs to provide more details. For example, when removing the coarse-to-fine component, what kind of policy is used as a replacement? In the "Ours w/o HLR" scenario, what kind of representation is used instead of the Holistic-Local Contrastive Representation for testing?
4. This paper aims to address task-level adaptation, but does this method also achieve similar improvements at the variation-level (though much easier but still important)?
5. Does this method demonstrate generalizability? For example, in the case of ML-10, if the training and test tasks were swapped, would the empirical performance still be as good?
6. It would be better if you could discuss the limitations of this work at the end of the paper.

**Details Of Ethics Concerns:**

None.

---

### Official Review · Reviewer_pDV9 · 2024-11-04

**Soundness:** 2
**Presentation:** 2
**Contribution:** 2
**Rating:** 5
**Confidence:** 4

**Summary:**

This research addresses limitations in meta-reinforcement learning for zero-shot adaptation, specifically for complex, task-level adaptation where agents encounter entirely new tasks, not just simple task variations. Traditional methods struggle due to weak task representations and inefficient policy learning, which don’t leverage the hierarchical structures of tasks.

To overcome these issues, the authors propose a Coarse-to-Fine Policy Refinement and a Holistic-Local Contrastive Representation approach. Initially, task language instructions are used to select skill-specific expert modules for a “coarse” policy. This is refined by a “fine” policy generated via a hypernetwork to produce task-aware actions. For task representation, a contrastive learning method is used to improve task understanding from both holistic and local perspectives.

**Strengths:**

The experimental results show that performance improvement is significant in Meta-World benchmark. I also believe the question the authors would like to solve in important in the community.

**Weaknesses:**

The proposed approach is somehow complicated, and the motivation for each design choice isn’t clearly explained. Please refer to the question section.

**Questions:**

- Is the expert policy jointly trained with the whole framework? If that’s the case, does the expert policy learn meaningful behavior? It’s likely that some of the expert policies were never used, so it learn nothing in the end.
    - Can the author provide more visualization for expert policy?
    - Is there any approach to evaluate the quality of the expert policy?
- I see the point of using coarse-to-fine policy refinement and holistic contrastive representation, but I am not quite sure of the rationale for combining them together.
    - I know the authors already provide the ablation study to empirically show the effectiveness of each components. However, it would be great if the authors can further provide insight about potential synergies between the coarse-to-fine policy and holistic-local representations
- Just out of curiosity, why didn’t the author compare the proposed approach with MAML [1], and PEARL [2] (some famous baselines in Meta-RL)?
    - Are there any specific reasons for excluding these well-known meta-RL methods?


[1] "Model-Agnostic Meta-Learning for Fast Adaptation of Deep Networks", Chelsea Finn, Pieter Abbeel, Sergey Levine, ICML 2017

[2] "Efficient Off-Policy Meta-Reinforcement Learning via Probabilistic Context Variables", Kate Rakelly, Aurick Zhou, Deirdre Quillen, Chelsea Finn, Sergey Levine, ICML 2019

---

### Official Review · Reviewer_ZAGG · 2024-11-06

**Soundness:** 3
**Presentation:** 3
**Contribution:** 2
**Rating:** 5
**Confidence:** 3

**Summary:**

This paper combines a Coarse-to-Fine Policy Refinement (CFO) and the Holistic-Local Contrastive Representation (HLR) method to handle both task-level adaptation and variation-level adaptation. The CFO part leverages language instructions to obtain a coarse policy from the Mixture-of-Expert module. The HLR part trains robust task representations to obtain a hypernetwork-based task-aware fine policy. Their experimental results on Meta-World show the algorithm's zero-shot adaptation abilities, and both CFO and HLR parts are essential for the improvement.

**Strengths:**

The main strength is that the contrastive representations embed the task information in a robust and interpretable way, as illustrated in Fig 3b. Experiments show that the learned representations contain hierarchical characteristics of non-parametric task variability. Combined with CFO, this mechanism boosts the performance of zero-shot adaptation to unseen tasks within just two episodes.

**Weaknesses:**

The proposed method requires access to the task categories labels and task IDs during training because the loss function $\mathcal{L}_{\text{LCR}}$ involves the category index $i$ and the task index $j$.

The proposed method depends on the pre-trained expert modules and HyperNetworks, which are not required in the baselines VariBAD, LDM, and SDVT. Therefore, to improve the fairness of the experiments, comparisons of the training time, number of parameters, and memory occupation are needed.

**Questions:**

Notations:

1. The definition of $x_{ijk}^-$ in Eq. $7$ and Eq. $8$ is confusing. From my perspective, in Eq. $7$, it is the $k$-th sample of learned task representation from category $j$, and $x_{ijk}^-$ does not depend on $i$. However, in Eq. $8$, it is the sample of task $k$ from category $i$, so $x_{ijk}^-$ does not depend on $j$ in Eq. $8$.

2. In Eq. $6$, $c_i^n$ is the $n$-th sample of task representations from category $i$. In Eq. $8$, $x_{ij}$ is the sample of task $j$ from category $i$. Can we say $c_i^j = x_{ij}$ ?

3. In Eq. $7$ and Eq. $8$, the samples $x_{ij}$ are the learned task representations from the encoder $q_{\phi}$. While $\mathcal{L}_{\text{VAE}}$ depends on both the encoder's parameter $\phi$ and the decoder's parameter $\theta$, do the HCR and LCR losses only depend on $\phi$?

4. In Algorithm 1 of Appendix A, the notation $n$ in $S_t = \{s_{t,1}, \cdots, s_{t,n}\}$ is not defined. Does $n$ equal the number of training tasks $K$?

Method:

1. What's the output of HyperNetworks? How does it affect the training of the fine policy?

2. In Figure 2, the coarse policy's output is an action $a_{\text{coarse}}$ sampled from $\pi_{\text{coarse}}(\cdot \mid s_t)$. Is $a_{\text{coarse}}$ the only input of the fine policy?

Experiments:

1. In Figure 4 (evaluation) and Figure 5 (training), both the units of the x-axis are $1e8$ steps. However, the evaluation of $n_{\text{roll}}=2$ episodes only has $500$ steps each. Why do Figure 4 and Figure 5 share the same unit of the x-axis?

2. The return of "Ours" on ML-45's Test tasks in Table 2 $(2911.7 ± 105.1)$ is larger than that in Table 1, Episode 2 $(2893.3 ± 107.5)$. However, the Test tasks' Success Rates of both ML-10 and ML-15 match those in Table 1, Episode 2 (ML-10:   $85.7 ± 4.9$, and ML-45:
  $71.4 ± 3.5$). What results in this discrepancy?

3. The baselines VariBAD, LDM, and SDVT do not contain the HyperNetworks. What's the performance of removing the HyperNetworks, i.e., directly feeding the learned HLR task representation $z$ to a fine policy with a fixed architecture? Can you add it to your ablation experiments?

---

### Note · Authors · 2024-11-22

**Comment:**

We appreciate the reviewers’ feedback and have chosen to withdraw our paper.

**Withdrawal Confirmation:**

I have read and agree with the venue's withdrawal policy on behalf of myself and my co-authors.